
# Global, high-resolution mapping of tropospheric ozone – explainable machine learning and impact of uncertainties

Clara Betancourt[1], Timo T. Stomberg[2], Ann-Kathrin Edrich[3], Ankit Patnala[1], Martin G. Schultz[1], Ribana Roscher[2,4], Julia Kowalski[5], and Scarlet Stadtler[1]

[1]Jülich Supercomputing Centre, Jülich Research Centre, Wilhelm-Johnen-Straße, 52425 Jülich, Germany
[2]Institute of Geodesy and Geoinformation, University of Bonn, Nußallee 17, 53115 Bonn, Germany
[3]Aachen Institute for Advanced Study in Computational Engineering Science (AICES), RWTH Aachen University, Schinkelstrasse 2a, 52056 Aachen, Germany
[4]Data Science in Earth Observation, Technical University of Munich, Lise-Meitner-Str. 9, 85521 Ottobrunn, Germany
[5]Methods for Model-based Development in Computational Engineering, RWTH Aachen University, Eilfschornsteinstr. 18, 52062 Aachen, Germany

**Correspondence:** Scarlet Stadtler (s.stadtler@fz-juelich.de)

**Abstract.** Tropospheric ozone is a toxic greenhouse gas with a highly variable spatial distribution which is challenging to map on a global scale. Here we present a data-driven ozone mapping workflow generating a transparent and reliable product. We map the global distribution of tropospheric ozone from sparse, irregularly placed measurement stations to a high-resolution regular grid using machine learning methods. The produced map contains the average tropospheric ozone concentration of the years 2010 - 2014 with a resolution of $0.1° \times 0.1°$. The machine learning model is trained on AQ-Bench, a precompiled benchmark dataset consisting of multi-year ground-based ozone measurements combined with an abundance of high-resolution geospatial data.

Going beyond standard mapping methods, this work focuses on two key aspects to increase the integrity of the produced map. Using explainable machine learning methods we ensure that the trained machine learning model is consistent with commonly accepted knowledge about tropospheric ozone. To assess the impact of data and model uncertainties on our ozone map, we show that the machine learning model is robust against typical fluctuations in ozone values and geospatial data. By inspecting the feature space, we ensure that the model is only applied in regions where it is reliable.

We provide a rationale for the tools we use to conduct a thorough global analysis. The methods presented here can thus be easily transferred to other mapping applications to ensure the transparency and reliability of the maps produced.

## 1 Introduction

Tropospheric ozone is a toxic trace gas and a short-lived climate forcer (Gaudel et al., 2018). Contrary to stratospheric ozone which protects humans and plants from ultraviolet radiation, tropospheric ozone causes substantial health impairments to humans when it enters the lung and destroys the lung tissue (Fleming et al., 2018). It is also the cause of major crop losses globally, as it damages plant cells and leads to reduced growth and seed production (Mills et al., 2018). Tropospheric ozone is a secondary pollutant with no direct sources, but with formation cycles depending on photochemistry and precursor emissions.



It is typically formed downwind of precursor sources from traffic, industry, vegetation, and agriculture, under the influence of solar radiation. Ozone patterns are also influenced by the local topography causing specific flow patterns (Monks et al., 2015; Brasseur et al., 1999). Depending on the on-site conditions, ozone can be destroyed in a matter of minutes or have a lifetime of several weeks with advection from source regions to remote areas (Wallace and Hobbs, 2006). The interrelation of these

factors of ozone formation, destruction, and transport is not fully understood (Schultz et al., 2017). This makes ozone both difficult to quantify and to control. See Brasseur et al. (1999) and Monks et al. (2015) for more details on the ozone life cycle. Public authorities recognize ozone-related problems. To quantify ozone, they install air quality monitoring networks (Schultz et al., 2015, 2017). Furthermore, they enforce maximum exposure rules to mitigate ozone health and vegetation impacts (e.g. European Union, 2008).

Tropospheric ozone research is currently seeing increased use of machine learning methods. Such "intelligent" algorithms can learn nonlinear relationships of ozone processes and connect them to environmental conditions, even if their interrelations are not well understood through process-oriented research. Kleinert et al. (2021) and Sayeed et al. (2021) used convolutional neural networks to forecast ozone at several hundred measurement stations, based on meteorological and air quality data. Large training datasets allowed them to train deep neural networks, resulting in a significant improvement over the first machine

learning attempts to predict ozone (Comrie, 1997; Cobourn et al., 2000). Machine learning is also extensively used to calibrate low-cost ozone monitors that can then complement existing ozone monitoring networks (Schmitz et al., 2021; Wang et al., 2021). Furthermore, costly chemical reactions schemes for ozone modeling in atmospheric models can be emulated using machine learning (Keller et al., 2017; Keller and Evans, 2019). Ozone and ozone precursor datasets which can be used as training data for machine learning models are being increasingly made available as FAIR (Wilkinson et al., 2016) and open

data. One of these datasets is AQ-Bench ('air quality benchmark dataset,' Betancourt et al., 2021b), a dataset for machine learning on global ozone metrics which also serves as training data for this mapping study.

We refer to mapping as a data-driven method for spatial predictions of environmental target variables. For mapping, a model is fitted to observations of the target variable at a number of measurement sites, which might even be sparse and of irregular placement. To fit the model, environmental features are used which are proxies for the target variable. A map of the target

variable is produced by applying the model to the spatially continuous features in the mapping domain. The history of such mapping methods in environmental applications starts with the identification of e.g. regional road salt contamination and traffic-related air pollution in the 1990s (Mattson and Godfrey, 1994; Briggs et al., 1997). For air pollution, it was deployed as an alternative to spatial interpolation and dispersion modeling which suffer from performance issues due to sparse measurements, and a lack of detailed source description (Briggs et al., 1997). In their 2008 review article, Hoek et al. describe these early

mapping studies as "linear models with little attention to mapping outside the study area". This has changed dramatically as nowadays the simple linear regression of the features is replaced by modern nonlinear machine learning algorithms which are often trained on thousands of samples (Petermann et al., 2021; Heuvelink et al., 2020). Mapping was shown to outperform other geostatistical methods such as Kriging in several studies (e.g. Li et al., 2019; Ren et al., 2020), and mapping domains are extended to the global scale (Lary et al., 2014; Bastin et al., 2019; Hoogen et al., 2019). More recently, it is questioned

whether machine learning is really the most suitable method to "map the world" (Meyer, 4 Mar 2020): Some studies may be





overconfident in their mapping results because they use inappropriate validation strategies (Meyer et al., 2018; Ploton et al., 2020). Doubts also arise when the mapping models are applied to areas that have completely different properties from the measurement locations (Meyer and Pebesma, 2021). Furthermore, uncertainty estimates of the produced maps are particularly important as they are often used as a basis for further research. The new approaches combined in this study improve uncertainty

issues, ensure explainability and applicability, and allow for a robust and consistent analysis of the machine learning results.

In this study, we produce the first fully data-driven global map of tropospheric ozone, aggregated in time over the years 2010-2014. This study builds upon Betancourt et al. (2021b) who proved that ozone metrics can be predicted using geospatial data. We do not only provide the map as a product, but also uncertainty estimates and explanations to ensure the trustworthiness of our results. We justify the choice of methods and clarify why they are necessary for a thorough global analysis. Sect. 2 contains

a description of the data and machine learning methods, including explainable machine learning and uncertainty estimation. Sect. 3 contains the results, which are discussed in Sect. 4. We conclude in Sect. 5.

## 2  Data and methods

### 2.1  Data description

Mapping with machine learning models requires two datasets: a dataset for training, testing, and validating the model, which

contains features and targets at the measurement sites, and a dataset for prediction, which contains only the features on a regular grid. In the following, we present the datasets used in this study. Additional technical details on these data and their sources are given in Appendix A.

#### 2.1.1  AQ-Bench dataset

We fit our machine learning model on the AQ-Bench dataset ('air quality benchmark dataset,' Betancourt et al., 2021b). The

AQ-Bench dataset is a machine learning benchmark dataset that was designed to relate ozone statistics at air quality measurement stations to easy-access geospatial data. It contains aggregated ozone statistics of the years 2010-2014 at 5577 stations all over the globe, compiled from the database of the Tropospheric Ozone Assessment report (TOAR, Schultz et al., 2017). The bulk of the stations is located in North America, Europe, and East Asia. The dataset contains different kinds of ozone statistics such as percentiles or health-related metrics. Of these statistics, this study solely focuses on the average ozone as target (Fig. 1).

The features in the AQ-Bench dataset characterize the measurement site and are proxies for ozone formation, destruction, and transport processes. For example, the 'altitude' and 'relative altitude' of the station are important proxies for local flow patterns and ozone sinks. Other features are 'population density' in different radii around every station, which are proxies for human activity and thus ozone precursor emissions. The full list of features and which ozone processes they are related to are documented by Betancourt et al. (2021b). The features we choose as candidates for this mapping study are listed in Table 1.

Features that are only available at station locations and not in gridded format are excluded because they cannot be used for





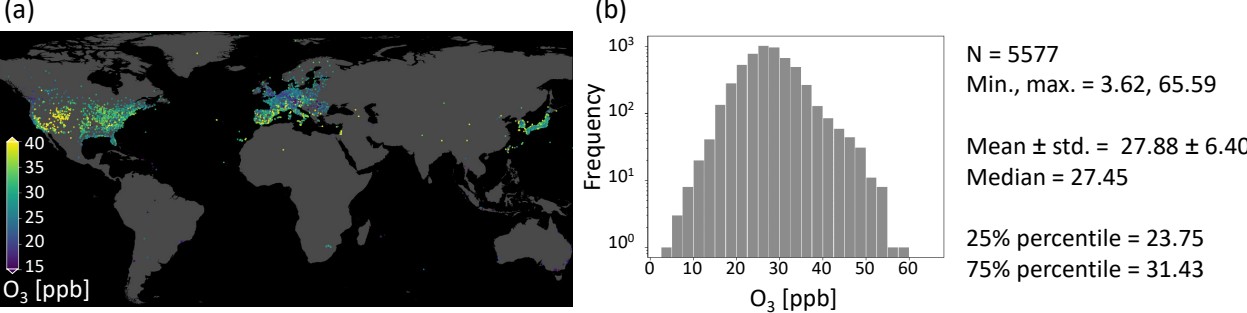

**Figure 1.** Average ozone values of the AQ-Bench dataset. (a) Values at measurement stations on a map projection. (b) Histogram and summary statistics.

mapping. With respect to geographical coordinates, only 'latitude' is used, which is a proxy for ozone formation through photochemistry. 'Longitude' is not a proxy for ozone formation.

### 2.1.2 Gridded data

To map the target average ozone, features are needed on a regular grid (i.e. as raster data) over the entire mapping domain.

Most of these gridded data are derived from the same original geospatial datasets as the features of the AQ-Bench dataset. The original data has a resolution of 0.1°×0.1° or finer. Since our target resolution is 0.1°×0.1°, the gridded data are downscaled to that resolution if the original resolution is finer. The 'land cover', 'population', and 'light pollution' features of the AQ-Bench dataset are not point data at the station, but spatial aggregates in a certain radius around the station (see Table 1). To prepare gridded fields of these features, the area around each individual grid point is considered, and the required

radius aggregation is written to that grid point. The gridded dataset is available under the DOI http://doi.org/10.23728/b2share. 9e88bc269c4f4dbc95b3c3b7f3e8512c. See Appendix A for details on the original data sources.

Table 1: Feature candidates selected from the AQ-Bench dataset.

|  | feature | Unit |
| --- | --- | --- |
| **General** | Climatic zone | - |
|  | Latitude | deg |
|  | Altitude | m |
|  | Relative altitude | m |
| **Land cover** | Water in 25 km area | % |
|  | Evergreen needle leaf forest in 25 km area | % |

*(continued on next page)*





*(Table 1 continued from previous page)*

|  | Feature | Unit |
|---|---|---|
|  | Evergreen broadleaf forest in 25 km area | % |
|  | Deciduous needle leaf forest in 25 km area | % |
|  | Deciduous broadleaf forest in 25 km area | % |
|  | Mixed forest in 25 km area | % |
|  | Closed shrublands in 25 km area | % |
|  | Open shrublands in 25 km area | % |
|  | Woody savannas in 25 km area | % |
|  | Savannas in 25 km area | % |
|  | Grasslands in 25 km area | % |
|  | Permanent wetlands in 25 km area | % |
|  | Croplands in 25 km area | % |
|  | Urban and built-up in 25 km area | % |
|  | Cropland / natural vegetation mosaic in 25 km area | % |
|  | Snow and ice in 25 km area | % |
|  | Barren or sparsely vegetated in 25 km area | % |
| **Agriculture** | Wheat production | 1000 tons $y^{-1}$ |
|  | Rice production | 1000 tons $y^{-1}$ |
| **Ozone precursors** | $NO_x$ emissions | g $m^{-2}$ $y^{-1}$ |
|  | $NO_2$ column | $10^5$ molec $cm^{-2}$ |
| **Population** | Population density | person $km^{-2}$ |
|  | Maximum population density in 5 km area | person $km^{-2}$ |
|  | Maximum population density 25 km area | person $km^{-2}$ |
| **Light pollution** | Nightlight 1 km | brightness index |
|  | Nightlight in 5 km area | brightness index |
|  | Maximum nightlight in 25 km area | brightness index |

## 2.2 Explainable machine learning workflow

We apply a standard mapping workflow and extend it with explainable machine learning methods as described in the following.

Together with the uncertainty assessment methods described in Sect. 2.3, they allow for a thorough analysis of the model. A random forest (Breiman, 2001) is fitted on the AQ-Bench dataset to output average ozone at the corresponding measurement stations for given features. A random forest is an ensemble of regression trees that is created by bootstrapping the training dataset several times to increase generalizability. We choose random forest as a machine learning algorithm because tree-based





models are the state of the art for structured data (Lundberg et al., 2020). Random forest was also shown to outperform linear
regression and a shallow neural network in predicting average ozone on the AQ-Bench dataset (Betancourt et al., 2021b).
In addition, this algorithm has been proven to be the most suitable for mapping in several studies (Petermann et al., 2021;
Nussbaum et al., 2018; Ren et al., 2020). We use the python machine learning framework SciKit-learn (Pedregosa et al., 2011).
We automate the hyperparameter search with the python package hyperactive (Blanke, 2021).

A proper validation strategy is crucial for spatial prediction models because both environmental conditions and target vari-
ables are often correlated in space. When tested on spatially correlated and thus statistically dependent samples, mapping
results may be overconfident (Meyer et al., 2018; Ploton et al., 2020). We use the independent spatial data split provided with
the AQ-Bench dataset to avoid this overconfidence. Details on our validation strategy are given in Sect. 2.2.1.

After training and validation, the model is applied point-wise to the gridded data with a resolution of $0.1° \times 0.1°$ to produce
the final ozone map. As an extension of this standard mapping workflow, we perform experiments to increase interpretability,
test robustness, and explain the model. The extended workflow is summarized in Table 2 and further justified in the following.

Table 2: Machine learning experiments as an addition to the standard mapping method. For details on the methods, please see
the given sections.

| Sect. | Method | Goal |
|---|---|---|
| **2.2.2** | Feature engineering | Make features easier to interpret |
| | Forward feature selection | Remove counterproductive features which favor overfitting |
| **2.2.3** | Spatial cross validation | Check model spatial robustness |
| | Cross validation on world regions | Evaluate model generalizability |
| **2.2.4** | Calculate SHAP values | Explain model predictions |

The use of redundant features in mapping applications can favor overfitting and even cause the machine learning model to
learn properties of individual locations. We thus remove counterproductive features by forward feature selection as proposed
by Meyer et al. (2018). Additionally, we apply basic feature engineering to increase the interpretability of the model. Details
on feature engineering and feature selection are described in Sect. 2.2.2.

In order to make our mapping model trustworthy, we need to verify its robustness and ability to generalize to previously
unseen locations, but also to explore the limits of its predictive capabilities. Noise in the AQ-Bench dataset might cause
problems if the model is not robust. Additionally, limited availability of ozone measurements in regions like Central and
South East Asia, Central and South America, and Africa is expected to pose a problem as is unclear whether our model will
generalize to these regions. Environmental factors and their interaction with ozone might be highly variable, especially over a
large domain such as the entire globe. Because of that, our model can have high evaluation scores when tested on the world
regions with many air quality measurement stations (Europe, North America, East Asia, see Fig. 1) but might not necessarily





be as reliable in other regions. To tackle the issues of robustness and generalizability we develop a spatial cross validation strategy in Sect. 2.2.3.

Finally, we aim to explain how the model arrives at its predictions, and to check if it is consistent with common ozone process understanding. For that, we use SHAP (SHapley Additive exPlanations, Lundberg and Lee, 2017), a post-hoc explainable machine learning method. It is a game-theoretic approach based on Shapley values (Shapley, 1953). In game theory, Shapely values provide a way of fairly distributing the outcome of a game among the 'players', the contributors to the game. For our random forest, they provide a means to identify the importance of the individual features to a model prediction. We describe

our SHAP implementation in Sect. 2.2.4.

### 2.2.1  Evaluation scores

We rely on the independent data split of AQ-Bench as provided by Betancourt et al. (2021b). Here, stations with a distance of more than 50 km are considered independent of each other. 60 % of the AQ-Bench dataset is used for training, and 20 % for validation. The remaining 20 % are only used for testing the final model that is used to generate the map.

The evaluation score is the coefficient of determination $R^2$,

$$R^2 = 1 - \frac{\sum_{m=1}^{M}(y_m - \hat{y}_m)^2}{\sum_{m=1}^{M}(y_m - \langle y \rangle)^2} \quad \text{with} \quad \langle y \rangle = \frac{1}{M}\sum_{m=1}^{M} y_m \tag{1}$$

where $m$ denotes a sample index, $M$ the total number of samples, $\hat{y}_m$ a predicted target value, and $y_m$ a reference target value. $R^2$ measures the proportion of variance in the output values that the model predicts. Thus, a larger $R^2$ represents a better model and the largest possible value is 1, which is equal to 100 %. To provide an additional evaluation score that directly indicates the

expected error of the predicted ozone levels, we also evaluate the root mean square error (RMSE) in ppb:

$$\text{RMSE} = \sqrt{\sum_{m=1}^{M} \frac{(y_m - \hat{y}_m)^2}{M}} \tag{2}$$

### 2.2.2  Feature engineering and feature selection

Basic feature engineering is performed to improve the interpretability of the model. Different types of savanna, shrublands, and forests are given individually in AQ-Bench (see Table 1). We merge them into 'savanna', 'forest', and 'shrubland' because

a high number of features with similar properties would make the model interpretation more difficult. Instead of 'latitude', we train on the 'absolute latitude', since radiation and temperature decrease when moving away from the equator, regardless of whether one moves south or north. The feature 'absolute latitude' thus has a direct meaning in regard to increased ozone formation favored through high radiation or temperature. Compared to experiments performed without feature engineering, we did not see any increase in evaluation scores on the validation set (not shown).

Our feature selection method follows Meyer et al. (2018), who propose to eliminate counterproductive features in spatial prediction models by forward feature selection. The model is initially trained on all possible 2-feature combinations. The combination with the highest evaluation score on the validation set is kept. The model is then trained on each remaining feature





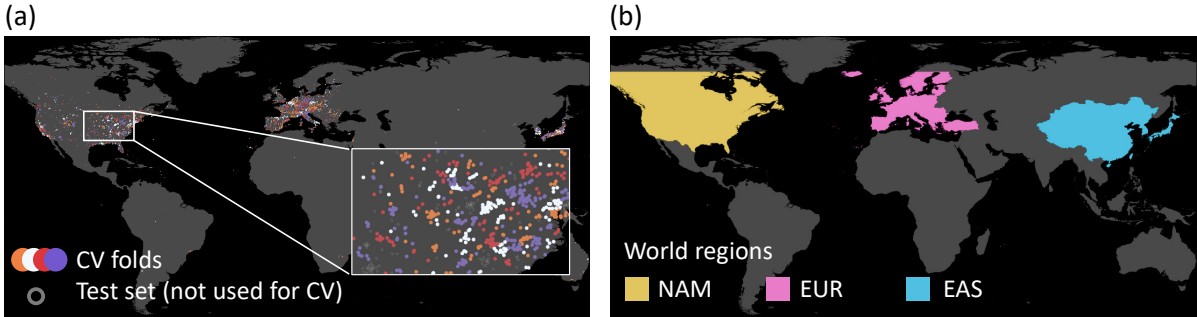

**Figure 2.** Data splits for the spatial cross validation. (a) Station clusters are randomly assigned to four cross validation (CV) folds. (b) The data is divided by the world regions North America (NAM), Europe (EUR), and East Asia (EAS).

along with the already selected features. The additional feature with the best evaluation score is appended to the existing list of features. This iterative approach is continued until the $R^2$ value drops, which indicates that a feature favors overfitting. The

final selected features are presented in Sect. 3.1.1.

### 2.2.3 Spatial cross validation

To prove a machine learning models' robustness, cross validation can be applied. We reserve 20 % of the AQ-Bench dataset for testing the final model, relying on the independent split of Betancourt et al. (2021b). We split the remaining 80 % into four independent cross validation folds of 20 % each. Like Betancourt et al. (2021b), we assume that air quality measurement

stations with a distance of at least 50 km are independent of each other. We, therefore, produce the cross validation folds with a two-step approach. First, we cluster the data based on the spatial location of the measurement sites using the density-based clustering algorithm DBSCAN (Ester et al., 1996). The maximum distance between clusters is set to 50 km so stations closer than that distance are assigned to the same cluster. Small clusters that result are randomly assigned our cross validation folds. In a second step, larger clusters (n > 50) are split again to ensure the same statistical distribution of all cross validation folds. For

this, we use the KMeans clustering algorithm (Duda et al., 2001). The resulting smaller clusters are again randomly assigned to the cross validation folds. Fig. 2 (a) shows this data split.

To evaluate the generalizability of our predictions to world regions with few measurements, we extend our spatial cross validation experiment. Here we divide the data by world region (Fig. 2 (b)). Most measurement sites are located either in North America, Europe, or East Asia, and we only consider stations in these world regions for this experiment. A random forest

is fitted and evaluated on two of the three regions and also evaluated on the third region for comparison. For example, it is fitted and evaluated on data of Europe and North America and additionally evaluated in East Asia. The difference in the resulting evaluation scores shows the spatial generalizability of the model. The results of the spatial cross validation experiments described in this section are presented in Sect. 3.1.2.





### 2.2.4 Shapley Additive Explanations (SHAP)

SHapley Additive exPlanations (SHAP) Lundberg and Lee (2017) provide detailed explanations for individual predictions by quantifying how each feature contributes to the result. The contribution refers to the average model output over the training dataset. In other words, this means that a feature with the SHAP value $x$ causes the model to predict $x$ more than the average prediction or base value (over the training set). To calculate SHAP for our data and model, we use the Python package SHAP provided by Lundberg (2021). The package contains a TreeSHAP module (Lundberg et al., 12 Feb 2018), which has been 185 specially tailored to tree-based models. It, therefore, provides an efficient and accurate approach for our random forest model.

Global feature importances are obtained by adding up all local contributions to the predictions. Features with high absolute contributions are considered more important. The local and global SHAP feature contributions aid us in checking for the scientific consistency of the model output. The SHAP values of our model are presented in Sect. 3.1.3.

### 2.3 Methods to assess the impact of uncertainties

Uncertainty assessment increases the trustworthiness of our machine learning approach and final ozone map. In general, the predictions of machine learning models have two kinds of uncertainties (Gawlikowski et al., 7 Jul 2021): First, model uncertainty, which results from the trained machine learning model itself, and second, data uncertainty which stems from the uncertainty inherent in the data. It is common to treat these uncertainties separately. Developing an uncertainty assessment strategy for our mapping approach is challenging because different uncertainties arise at different stages of the mapping process. Looking at it 195 closely, every ozone measurement, every preprocessing step, and every model prediction is a potential source of error. It would be infeasible to investigate the impacts of each and every error. We, therefore, identify the most important error sources and analyze the uncertainty induced in our produced map only for these. The decision on which aspects to analyze specifically is based on expert knowledge and on the results of our machine learning experiments, i.e., robustness analysis (Sect. 2.2.3) and SHAP values (Sect. 2.2.4). We develop a formalized approach which is summarized in Table 3 and further elaborated in the 200 following.

Table 3: Uncertainty assessment for our mapping method. For details on the methods, please see the given sections.

| Sect. | Method | Goal |
|---|---|---|
| **2.3.1** | Define area of applicability | Ensure the model is only applied where it is reliable |
| **2.3.2** | Modeling of ozone fluctuations | Evaluate the impact of ozone fluctuations on produced map |
| **2.3.3** | Propagate subgrid altitude variation through model | Evaluate uncertainty introduced by altitude variation |

The model error is caused by the uncertainty of the trainable parameters of a machine learning model. Uncertainties in the model can become visible, for example, when different model results are obtained if the model is initialized with different random seeds before training (as for example in Petermann et al., 2021). To rule out this training instability, we re-trained





our models several times with different random seeds and monitored the results. We have seen negligible variations and thus rule out this kind of uncertainty (not shown). Apart from uncertainty through training instability, the model uncertainty is usually also high for predictions in areas of the feature space where training data is sparse (Lee et al., 26 Nov 2017; Meyer and Pebesma, 2021). For example, a model that was not trained on data from very high mountains or deserts is not expected to produce reliable results in areas with these characteristics. For this reason, we apply the concept of 'area of applicability'

by Meyer and Pebesma (2021) to limit our mapping to regions where our model is expected to produce reliable results. The details are described in Sect. 2.3.1.

  Of the data errors, the error caused by the target variable 'average ozone' is the first choice for assessment. Fluctuations and random measurement errors introduce uncertainty into the ozone measurements. We evaluate the uncertainty caused by these influences using a simple error model. To see the influence of ozone fluctuations on the final map, the error model is used to

perturb the training data – and we can check how the final map changes when trained on perturbed instead of original data. The error model is described in Sect. 2.3.2.

  Additional data uncertainty stems from the features. For example, geospatial data derived from satellite products are sensitive to retrieval errors. Based on the sources and documentation of our geospatial data (Appendix A), we expect such errors to have a small impact in this study. However, we want to take a closer look at subgrid features in the geospatial data, and how they

affect the model results. We limit ourselves to the 'altitude' because our SHAP analysis (Sect. 3.1.3) has shown that it is the most important feature besides 'latitude' which does not have critical subgrid variations. Subgrid variations of the altitude might influence our final map, especially if a feature like a cliff or a high mountain is present in the respective grid cell. We evaluate the influence of subgrid variations in height on the final map by propagating higher resolution altitudes through the final model as described in Sect. 2.3.2.

**2.3.1 Area of applicability method**

We adopted the area of applicability method from Meyer and Pebesma (2021), and we refer to that study for a detailed derivation. The method is based on considering the distance of a prediction sample to training samples in the multidimensional feature space. This concept is illustrated in Fig. 3, where it can be clearly seen that the AQ-Bench dataset forms a cluster in the feature space, but that our mapping domain contains feature combinations that do not belong to this cluster. Predictions made

on these feature combinations suffer from high uncertainty. Consequently, we use the area of applicability method to flag data points with a great distance to the training data cluster as 'not predictable'.

  The features are first normalized to treat differences in all features equally. Second, the features are scaled according to the feature importance to make distances of important features more relevant. For this importance scaling, we reuse the global feature importances as provided by SHAP (Sect. 3.1.3). To find a threshold distance for non-predictable samples, we rely again

on our cross validation sets described in Sect. 2.2.3. In more detail, we consider the distance from every training data point to the closest data point in a different cross validation set. The threshold distance for 'non-predictable' data is the upper whisker of all the cross validation distances. Since the model is trained on land surface data only, we also remove the oceans from the area of applicability. The result of this experiment is shown in Sect. 3.2.1.





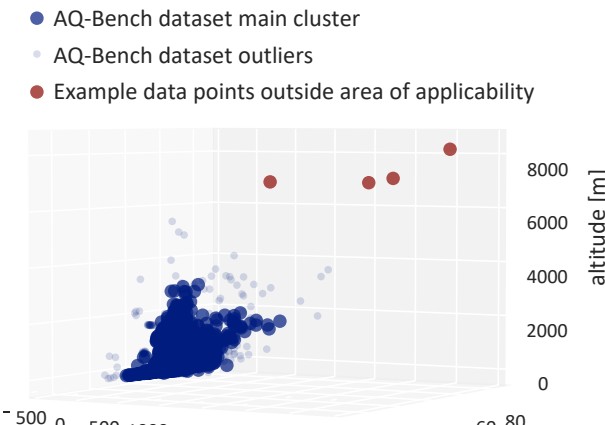

**Figure 3.** Principle of the area of applicability. The plot displays the distribution of all AQ-Bench samples along the three most important feature axes 'absolute latitude', 'altitude', and 'relative altitude'. It is clearly visible that the AQ-Bench samples form a cluster, and that some feature combinations in the gridded data are far away from that cluster.

### 2.3.2 Modeling ozone fluctuations

Here we describe our error model for evaluating the uncertainty introduced by typical ozone biases in the produced maps. Such biases may arise from measurement uncertainties, local geographic effects, or an "unusual" environment with respect to precursor emission sources. We consider all of these effects as ozone measurement uncertainties although it would be more precise to say that they are uncertainties in the determination of ozone concentrations at the scale of our grid boxes.

Quantification of these uncertainties is challenging, as we typically lack the necessary local information. Here, we assume
the local ozone values to be subject to a Gaussian error with mean 0 ppb and variance 5 ppb as suggested by the discussion on ozone data errors in Schultz et al. (2017), Sect. 4. The principle of the error model is to randomly perturb a subset of the training ozone values according to typical uncertainties of ozone measurements and to monitor resulting variances in the final map. Assuming only one-quarter of the measurement values are biased, a random subset consisting of 25 % of the training ozone values is either increased or decreased by random values sampled from a Gaussian distribution with 0 ppb mean and
5 ppb variance. We use multiple realizations of this error model to perturb the training data, each realization perturbing a different subset with different values. One example error model realization is shown in Fig. 4.

The principle of propagating the ozone error and to analyze its impact on the resulting map is then to train on the randomly perturbed data, obtain a 'perturbed model', and then create 'perturbed maps'. If the perturbations of the resulting ozone maps are less or equal to the initial perturbations, the resulting uncertainty in the map is considered acceptable. If completely different
maps would be produced, this would point to a model lacking robustness. The process of perturbing, training, and comparing

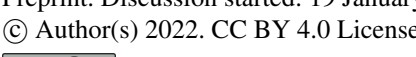



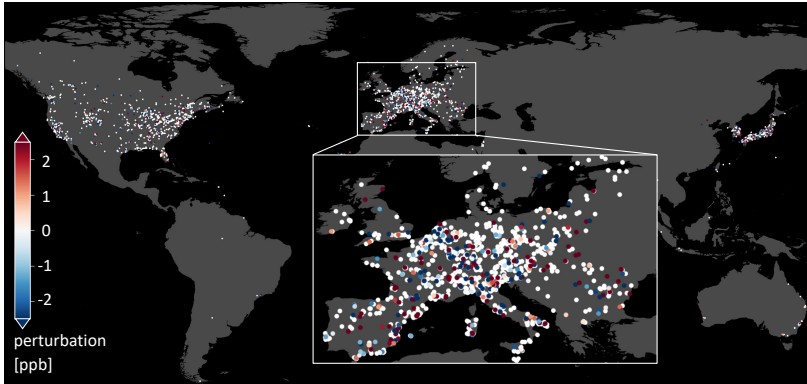

**Figure 4.** Example realization of the error model for ozone uncertainties. A random subset of 25 % of the ozone values in the training set is perturbed with values sampled from a Gaussian distribution with 0 ppb mean and 5 ppb variance.

maps is repeated until the standard deviation of all perturbed maps converges. For the configuration considered in this study, the error model converged fully after 100 realizations, see Appendix C for details and further justification. The result of this experiment is presented in Sect. 3.2.2.

### 2.3.3 Propagating subgrid altitude variation through model

In contrast to perturbing the targets and retraining the machine learning model, here we sample inputs from a finer resolution grid and propagate them through the existing fitted model. In more detail, for every grid cell of our final map with 0.1° resolution, we propagate all 'altitude' values of the original finer resolution digital elevation model (DEM, resolution 1′, see Appendix A) through our random forest model while leaving the other variables unchanged. For each coarse 0.1° resolution grid cell we find 36 altitude values of the fine grid cells and can thus make 36 predictions. We monitor the deviation of these

predictions from the reference prediction in that cell. The results of these experiments are presented in Sect. 3.2.3.

### 3   Results

The results of our explainable machine learning mapping workflow (Sect. 2.2, Table 2) are presented in Sect. 3.1. The impact of uncertainties (Sect. 2.3, Table 3) are presented in Sect. 3.2. The final ozone map that is generated based on the knowledge gained from all experiments is presented in Sect. 3.3.





### 3.1 Explainable machine learning model

#### 3.1.1 Selected hyperparameters and features

We choose the following standard hyperparameters for our random forest model: 100 trees are fitted on bootstrapped versions of the AQ-Bench dataset with a Mean Square Error (MSE) loss function and unlimited depth. The evaluation scores of our random forest proved not to be sensitive to the choice of hyperparameters (not shown). Therefore, the standard hyperparameters are used to fit the model in all experiments of this study.

Based on the forward feature selection (Sect. 2.2.2) the following variables are used to build the model:

- Climatic zone
- Absolute latitude
- Altitude
- Relative altitude
- Water in 25 km area
- Forest in 25 km area
- Shrublands in 25 km area
- Savannas in 25 km area
- Grasslands in 25 km area
- Permanent wetlands in 25 km area
- Croplands in 25 km area
- Rice production
- $NO_x$ emissions
- $NO_2$ column
- Population density
- Maximum population density in 5 km area
- Maximum population density 25 km area
- Nightlight 1 km
- Nightlight in 5 km area
- Maximum nightlight in 25 km area

The following features are discarded because the validation $R^2$ score decreases when they are added to train the model: 'Urban and built-up in 25 km area', 'cropland / natural vegetation mosaic in 25 km area', 'snow and ice in 25 km area', 'barren or sparsely vegetated in 25 km area', 'wheat production'. A discussion of why these features might be counterproductive follows in Sect. 4.1.

#### 3.1.2 Spatial cross validation reveals limits in the model generalizability

The four-fold cross validation from Sect. 2.2.3 results in $R^2$ values in the range of 0.58 to 0.64 and RMSEs in the range of 3.83 to 4.04 ppb (Table 4). These evaluation scores show that all models are useful despite the spread in evaluation scores. On average the models explain 61 % of the variance in ozone values. The mean RMSE is 3.97 ppb. To put this RMSE value into perspective, 5 ppb is a conservative estimate for the ozone measurement error (Schultz et al., 2017). Although the evaluation scores of all folds are in an acceptable range, the standard deviation of 0.08 ppb in the RMSEs shows that evaluation scores depend to some extent on the data split.





Concerning the spatial cross validation on different world regions, the $R^2$ value drops between 0.13 and 0.49 when training and validating in different world regions (Table 5). The RMSEs increase when training and validating in different world regions

with the exception of the East Asia test case where the RMSE barely changes. Regarding the evaluation scores, East Asia is a special case because the ozone value distribution is rather narrow there (not shown). This explains the low $R^2$ value and the acceptable RMSE. One reason for the change in evaluation scores when training and testing in different world regions could be very different feature combinations of the different world regions. We have ruled out this reason by inspecting the feature space (similar to Sect. 2.3.1, not shown). The only other possible reason for the decrease in $R^2$ is that the relationship between

features and ozone is not the same in different world regions. Therefore, the expected evaluation scores of our map vary not only with the feature combinations (as described in Sect. 2.3.1), but also spatially. We differentiate between the two issues and their influence on the model applicability in Sect. 3.2.1 and discuss them further in Sect. 4.3.

**Table 4.** Four-fold cross validation results.

| Fold | $R^2$ | RMSE [ppb] |
|---|---|---|
| 1 | 0.64 | 3.83 |
| 2 | 0.58 | 4.03 |
| 3 | 0.61 | 4.04 |
| 4 | 0.61 | 3.97 |
| ∅ | $0.61 \pm 0.02$ | $3.97 \pm 0.08$ |

**Table 5.** Cross validation on the world regions Europe (EUR), East Asia (EAS), and North America (NAM). We also give the difference in $R^2$ values and RMSEs, when validating the model in another world region than the training region.

| Training region | Validation region | | $R^2$ | RMSE [ppb] |
|---|---|---|---|---|
| EUR + EAS | EUR + EAS | | 0.57 | 3.54 |
| | NAM | | 0.34 | 5.01 |
| | | *diff.* | - 0.23 | + 1.47 |
| EAS + NAM | EAS + NAM | | 0.52 | 3.76 |
| | EUR | | 0.39 | 4.64 |
| | | *diff.* | - 0.13 | + 0.88 |
| NAM + EUR | NAM + EUR | | 0.63 | 3.92 |
| | EAS | | 0.14 | 3.78 |
| | | *diff.* | - 0.49 | - 0.14 |

### 3.1.3 SHAP values quantify the influence of the features on the model results

SHAP was used to determine the feature importance of the random forest model as described in Sect. 2.2.4. Fig. 5 contains a

summary plot with the global feature importance (left side) and SHAP values of all features on the test set (right side). The global importance of the features 'absolute latitude', 'altitude', 'relative altitude', and 'nightlight in 5km area' are highest with a contribution of at least 10 %. The remaining features have a weaker influence on the model output. E.g. the influence of the 'climatic zone' is often negligible. The local SHAP values in Fig. 5 reveal the contribution of features to the predictions. Here it



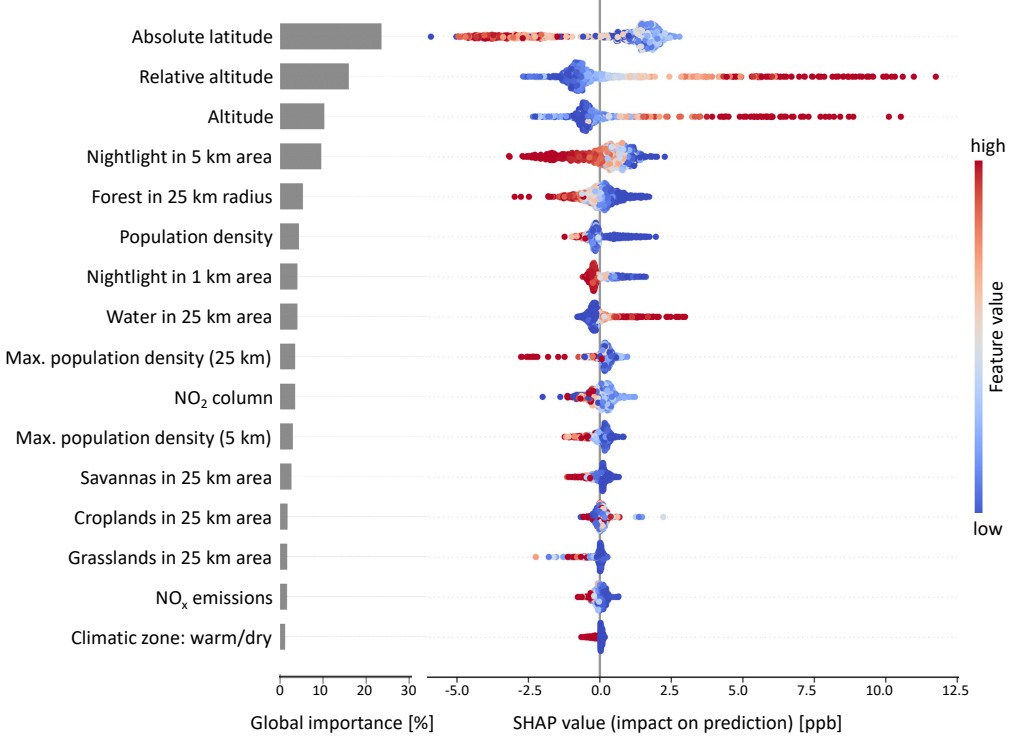

**Figure 5.** SHAP summary plot. The global importances on the left side are calculated from the averaged sum of the absolute SHAP values. The dots in the beeswarm plots on the right side show the SHAP values of single predictions. The color indicates the respective feature value. This plot shows only features with more than 1 % global importance.

can be seen, for example, that a low 'absolute latitude' value, i.e., a location near the equator, leads to an increased ozone value
prediction. Likewise higher 'altitude' and 'relative altitude' increase predicted ozone values. Very high 'nightlight in 5km area' values lead to lower predicted ozone concentrations. These tendencies are in line with domain knowledge on the atmospheric chemistry of ozone. We discuss the physical consistency of the model based on the SHAP values in Sect. 4.1.

Fig. 6 shows two specific examples of accurate (less than 1 ppb error) predictions for a low-ozone and a high-ozone station, respectively. The high ozone station (Fig. 6 (a)) is located in a rural area in the US with many agricultural fields and a smaller
city nearby. The average ozone at this location is predicted to be high because the model uses the absence of forests, the low 'night light in 5 km area' value, and the 'absolute latitude' as features leading to high ozone values. This is consistent with Fig. 5 where it can be seen that a lower 'absolute latitude' often increases the ozone value. The French station (Fig. 6 (b)) is an urban background station surrounded by fields. The location is further in the north than the US station which leads to a strong decrease in the predicted ozone value. The low '(relative) altitude' further decreases the predicted ozone.





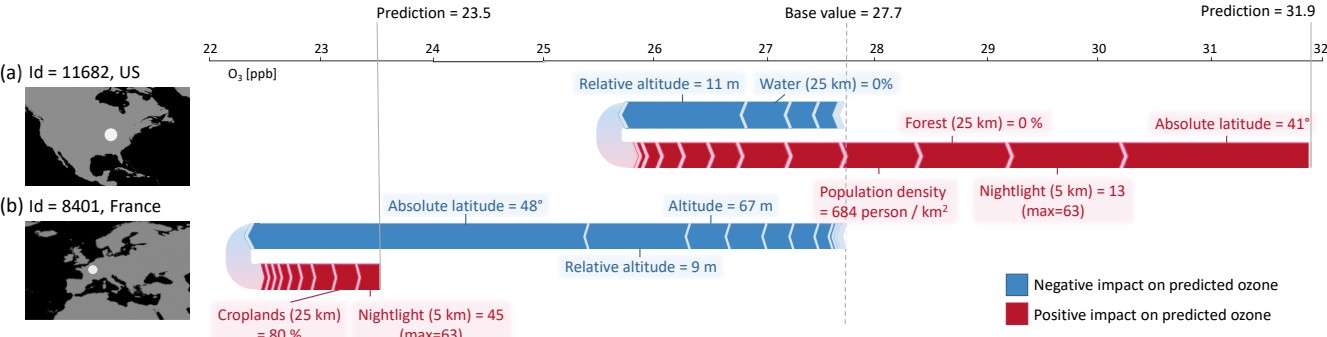

**Figure 6.** SHAP force plots for two example predictions at a) a rural station in the US and (b) an urban station in France. Starting from the base value (27.7 ppb) which is the mean of all predictions, a feature can increase or decrease the predicted ozone (red and blue arrows). The final predictions (23.5 and 31.9 ppb respectively) result from adding all SHAP values to the base value. The most contributing features are labeled and their values are given.

## 3.2 Evaluating the impact of uncertainties

### 3.2.1 Applicability and uncertainty of the model depends on both features and location

As described in Sect. 2.3.1, predictions of our model are considered valid if the feature combinations are similar to those of the training dataset AQ-Bench. Additionally, the results of the spatial cross validation (Sect. 3.1.2) have shown that the spatial proximity to the training locations has an influence on the model performance and uncertainty. Two cases were examined in this section: Firstly, the cross validation sets which are close to each other (RMSE in the range of 0.4 ppb, as seen in Table 4), and secondly, the cross validation on different world regions, that have a maximum distance from each other (RMSE values of up to 0.55 ppb, as seen in Table 5). In our uncertainty assessment, we combine findings from both the area of applicability (for matching features) and the spatial cross validation methods (for spatial proximity). The criterion for matching features was presented in Sect. 3.2.1. To make a distinction for spatial proximity, we consider the mean spatial distance of a measurement station to the closest measurement station in another cross validation set. It is ca. 182 km.

Fig. 7 shows the area of applicability of our model including this spatial distinction. In this figure, locations with unrestricted applicability (matching features and spatial proximity of up to 182 km to training locations) are marked in bright turquoise. Here we expect an RMSE in the range of 4 ppb and an $R^2$ value of about 0.55. We mark locations with a spatial distance of more than 182 km from training locations in a darker shade of turquoise. Here the RMSE may raise to about 5 ppb. Bright grey areas in Fig. 7 denote areas that are closer than 182 km to a measurement station but do not have feature combinations found in AQ-Bench.

The bulk of the regions with good coverage of measurement stations (North America, Europe, and parts of East Asia) are well predictable. In these regions, only some areas high in the north and high mountains are not predictable. Conversely, large areas in South and Central America, Africa, far northern regions, and Oceania have feature combinations different from the





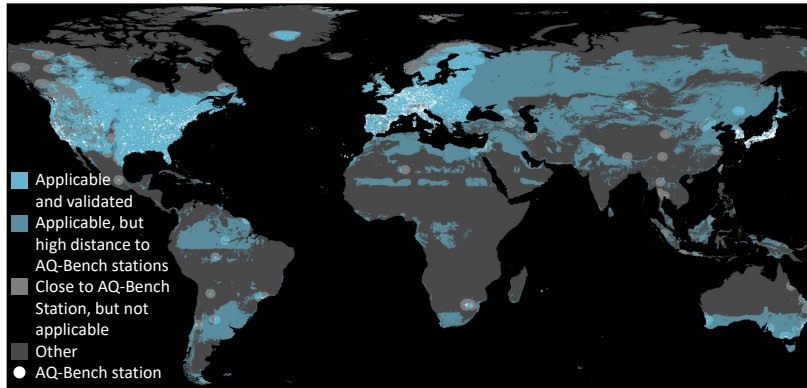

**Figure 7.** Area of applicability with restrictions in the feature space and spatial restrictions. The bright turquoise areas fulfill all prerequisites to be predictable: they have similar features as the AQ-Bench dataset and they are close to stations for validation. The darker shade of turquoise indicates similar predictions, but no proximity to stations for validation. Light grey areas indicate the proximity of a station, but no applicability of the model. The locations of all measurement stations are plotted in white.

training data and are therefore not predictable. There are some regions in the Baltic area, South America, Africa, and South Australia where feature combinations can be predicted by the model, but they are far away from the AQ-Bench stations. A broader discussion of the global applicability of our machine learning model follows in Sect. 4.3.

### 3.2.2 Uncertainty due to ozone fluctuations is within an acceptable range

The error model for ozone uncertainties is described in Sect. 2.3.2. The error model converged fully after 100 realizations, see
Appendix C for details on the error model convergence. The $R^2$ values of the perturbed models varied between 0.50 and 0.58. Fig. 8 shows the resulting standard deviation in the mapped ozone. We find that the assumed ozone fluctuations may lead to a less certain prediction in specific areas, such as areas with sparse training data. In general, it can be concluded, however, that our error model does not tend to amplify the effects of perturbed training data. This means that the machine learning algorithm smoothes out noise during training. This can be explained by the core functioning of the random forest which uses
bootstrapping during training.

Fig. 8 also shows that regions with poor spatial coverage by measurement stations (darker shade of turquoise in Fig. 7) are more sensitive to noisy training data. Example regions are the patches in Greenland, Africa, Australia, and South America. This can be explained by the fact that the model relies its predictions on a few samples and is thus very sensitive to perturbations of these few measurements.

### 3.2.3 Uncertainty through subgrid DEM variation is within an acceptable range

This method was described in Sect. 2.3.3. In most regions of the world, subgrid DEM variations around mean altitude are below 50 m (Fig. 9 (a)), e.g., in the central and eastern United States and in Europe except for the Alps. There are regions with



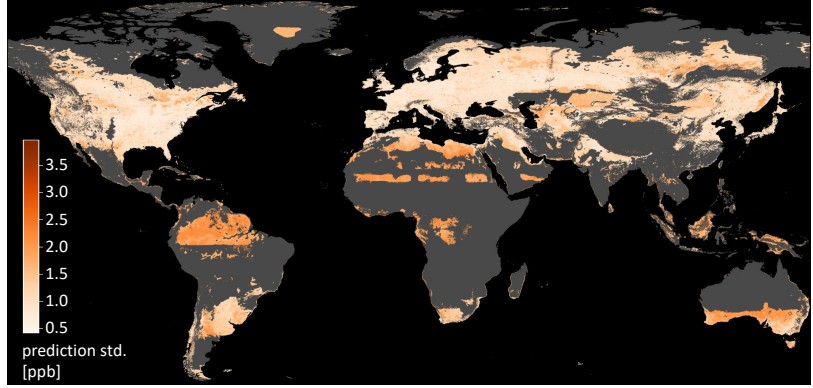

**Figure 8.** Standard deviation of the ozone predictions under perturbations. This map was created by stacking the maps of 100 error model realizations along the z axis and then calculating the grid point-wise standard deviation along the z axis.

higher variances such as the Rocky Mountains and their surroundings, the Alps, and large parts of Japan outside Tokyo. In Figure 9 (b) it can be seen how these variations influence the predicted ozone values. In the flat regions, the variance is below

0.5 ppb, and even in the very high variance regions, the deviation is very seldomly above 2 ppb. This means the model is robust against these variances. Few exceptions are present at the border of the area of applicability (ref. Sect. 3.2.1), e.g. in the Alps. But even in these regions, the deviation is well below 5 ppb, which is a conservative estimate for ozone fluctuations (Schultz et al., 2017). A discussion of implications for general subgrid variances can be found in Sect. 4.1.

### 3.3 The final ozone map

#### 3.3.1 Production of the final map

All selected features listed in Sect. 3.1.1 are used to fit the final model. In contrast to the experiments in the previous sections, we now train the model on 80 % of the AQ-Bench data set and test it on the remaining 20 % of the independent test set. Fig. 10 shows the predictions of this model on the test set vs. the true average ozone values. The $R^2$ value of this model is 0.55 and the RMSE is 4.4 ppb. Not all points are exactly on the 1:1 line, but there is a spread around it. Furthermore, true values of less

than 20 ppb or more than 40 ppb are predicted with high bias, which is expected since random forests tend to predict both low and high extremes less accurately than values closer to the mean.

#### 3.3.2 Visual analysis

The final map is shown in Fig.11 (data available under the DOI http://doi.org/10.23728/b2share.a05f33b5527f408a99faeaeea033fcdc). Predictions in the area of applicability are in a range between 9.4 and 56.5 ppb. This is not the full range of measured ozone

values (Fig. 1). There are some characteristics that are visible at first sight, e.g. the north-south gradient in Europe and generally higher values in mountain areas, like in the western US. Sometimes the borders of climatic zones lead to steps in the mapped



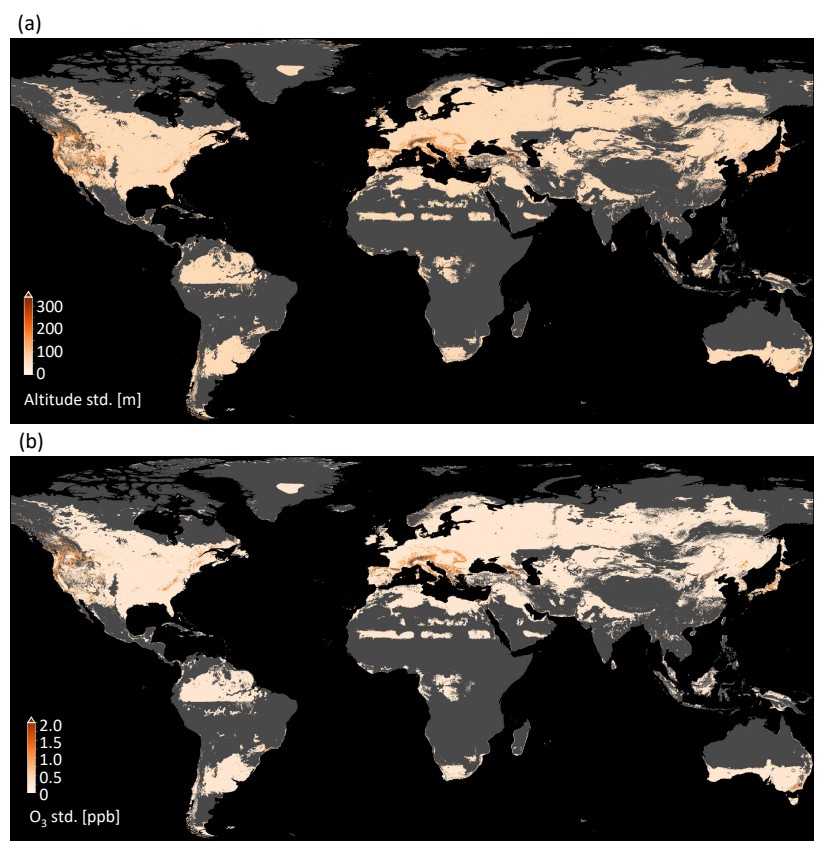

**Figure 9.** Results of propagating subgrid DEM variations through the model. (a) Spread of subgrid Digital elevation model data. (b) Spread of ozone values.

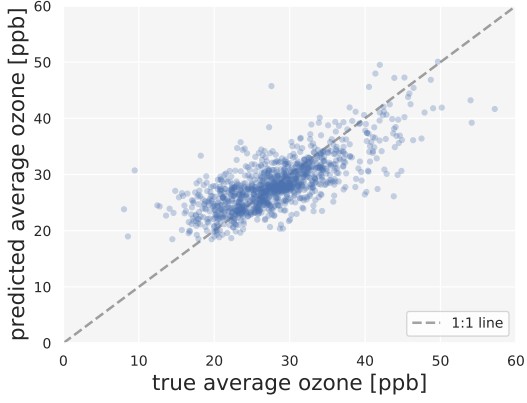

**Figure 10.** Predicted ozone values versus measurement values of the test set. There is a spread around the 1:1 line, furthermore, extremes are not captured as well as values closer to the mean.



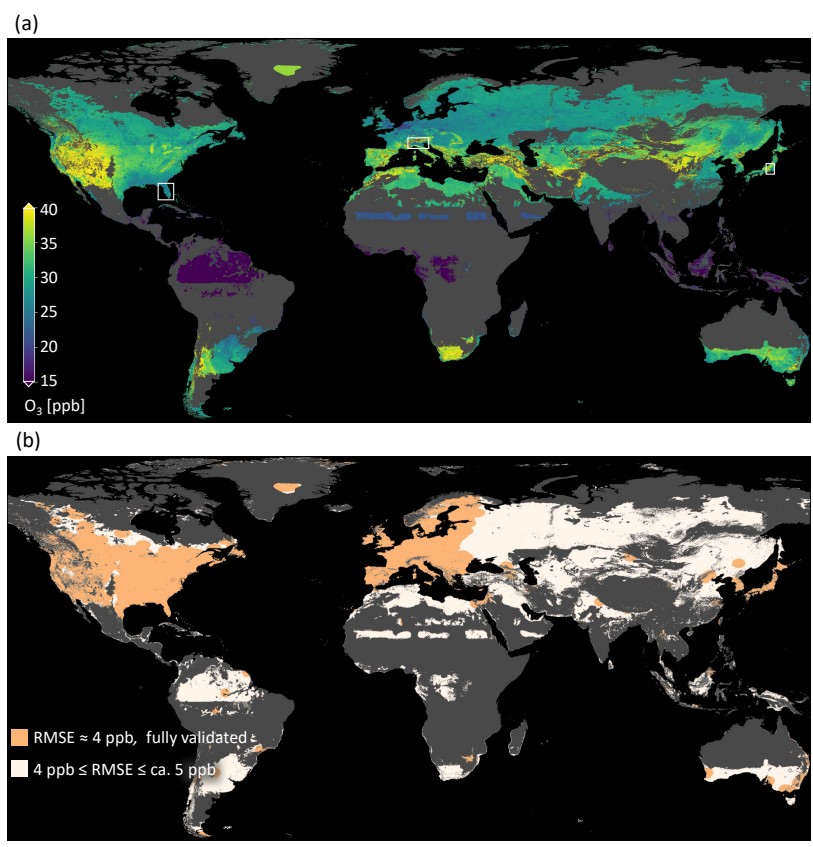

**Figure 11.** The final ozone map as produced in this study. The areas shown in Fig. 12 are highlighted by white boxes.

ozone values, like in the north of North America, and in Asia. This shows that even if the climatic zones are not important globally, they can be important locally. There are furthermore larger areas with very low ozone variation in Greenland, Africa, and South America.

In Fig. 12, a detailed look at three selected areas is given, and the predictions are compared to the true values. In image (a), a uniform, low ozone concentration is predicted over the peninsula of Florida. Image (b) shows low ozone values in the Po valley, a plane where many people live. Towards the mountains which surround the valley, higher values are predicted, and for the higher mountains, no predictions can be made. In image (c) we can see the city of Tokyo which is very well covered with ozone measurements and where ozone values are relatively low. Also at the coasts of Japan, the values are lower. Conversely on the
mountains, just as in image (b), higher levels of ozone are predicted and some areas can not be predicted. The spatial ozone patterns described here can also be found in ozone model products such as the fusion products by DeLang et al. (2021). We discuss the prospects of global ozone mapping more thoroughly in Sect. 4.4.





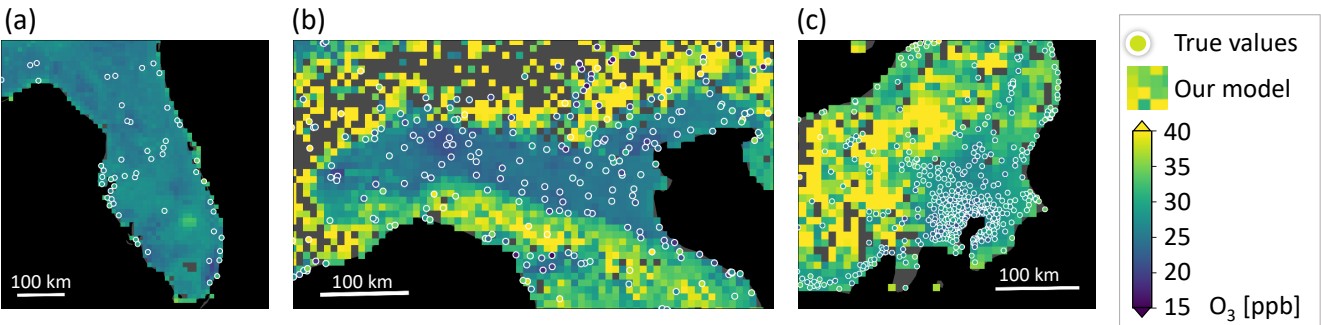

**Figure 12.** Map details with true values are given as white circles. (a) The Florida peninsula, US. (b) The Po Valley in northern Italy. (c) Tokyo, Japan, and its surroundings.

## 4 Discussion

### 4.1 Robustness

Based on Hamon et al. (2020), we define robustness in the context of this work as follows: *The model and map are considered robust if they do not change substantially under noise or perturbations that could realistically occur.* Here, 5 ppb change in RMSE score and the ozone map are considered significant, because 5 ppb are a conservative estimate for ozone measurement errors (Schultz et al., 2017).

Methods to assess the robustness are part of both the explainable machine learning workflow (Table 2) and the uncertainty 410 assessments (Table 3) of this study. Regarding the robustness of the training process, the cross validation results in Table 4 show that the model performance depended on the data split, leading to variances of the RMSE between 3.83 and 4.04 ppb. This was already noted by Betancourt et al. (2021b) and is regarded as an inherent limitation of a relatively small dataset. Apart from that, there were no robustness issues of the training process, e.g. evaluation scores did not vary with different random seeds (not shown). We tested also the robustness regarding typical variances in the ozone and geospatial data. The results from 415 Sect. 3.2.2 and Sect. 3.2.3 show that the produced ozone map is robust against these fluctuations. The variances are never above the initial perturbations, and variances in the map do not exceed the limit of 5 ppb defined above. Limits in the robustness were only shown through variances above 3 ppb at the borders of the area of applicability of the model, and in regions with sparse training data (grey and dark turquoise areas in Fig. 8 and 9). This outcome is especially interesting because it shows that the issues of applicability (discussed in more detail in Sect. 4.3) and robustness are interconnected. In areas where the model is 420 applicable, it is also more robust and uncertainties are lower.

In order to make the robustness assessment with respect to data feasible, we strongly reduced the dimensionality of our error model by using expert knowledge about the problem. We only conducted two experiments where we modify training data and model inputs (described in Sect. 2.3.2 and 2.3.3). These experimental setups were chosen because they are expected





to generalize well to other similar experiments. Firstly, spatial fluctuations produce similar perturbations to temporal fluc-
tuations (not shown), and, secondly, subgrid variances of one feature are also expected to generalize to other features. The
combined robustness experiments have shown that our produced maps are robust.

## 4.2   Scientific consistency

Here we discuss the scientific consistency of our model by assessing the results of the explainable machine learning work-
flow (Table 2). In more detail, we interpret the selected features, their importance, and their influence on the model predictions.
In our case, the features are proxies to ozone processes, which makes it challenging to interpret the underlying chemical pro-
cesses. Nevertheless, the connections between the features can be discussed, if they are plausible and consistent with respect to
our understanding of ozone processes. This is a pure a posteriori approach, meaning we did not in any way enforce scientific
consistency in the model or training process.

Regarding the global feature importance of SHAP (Fig. 5), it might, at first sight, be counterintuitive that the model focuses
more on geographical features such as 'absolute latitude' and 'altitude' than chemical factors such as the 'NO$_2$ column', 'NO$_x$
emissions'. Geographic features are proxies for flow patterns and heat, not for ozone chemistry, which ozone researchers
would expect to be of greater importance. This contradiction is due to the fact that the model provides an as-is view of ozone
concentration and is not process-oriented in any way. Please note that many features such as 'nightlight' and 'population
density' are correlated, so retraining the model might swap dependence in the SHAP values as noted by Lundberg et al. (2020).
The beeswarm plot in Fig. 5 aids in checking the physical consistency of our model. The effect of 'absolute latitude' on
predictions is consistent with what is known about ozone formation – ozone generally increases toward the equator. This is
also evident in the highly latitudinally stratified ozone overview plots in global measurement-based overview studies such
as *TOAR health* and *TOAR vegetation* (Fleming et al., 2018; Mills et al., 2018). High 'relative altitude' and 'altitude' both
increase the model ozone. This altitude-ozone relation is consistent with our previous knowledge (Chevalier et al., 2007).
There are also a few relatively important chemistry-related features. We can see that very high 'nightlight in 5 km area' reduces
the model ozone. This is consistent with NO titration (Monks et al., 2015). Nightlights are a proxy for human activity, generally
in the context of fossil fuel combustion, which usually leads to elevated NO$_x$ concentrations. NO reacts with ozone and thus
removes it, especially during the night time this leads to very low ozone levels. Conversely, very high 'forests in 25 km area'
values lead to lower ozone predictions. This is plausible because there is little human activity in forested areas and thus no
combustion-related precursor emissions occur. Quantification of either influence is not possible because, for example, it is
unclear to what extent the different forests emit volatile organic compounds which are also ozone precursors, and a city with
'nightlight in 5 km area' = 50 cannot be directly quantified in terms of precursor emissions either.

SHAP values also offer the possibility to quantify the influence of features on single predictions (Fig. 6). This is helpful
for certain special cases, e.g., when only a single prediction needs to be explained. In a global application, however, it might
become infeasible – not all the pixels in our ozone map can be explained one by one.

The forward feature selection (Sect. 2.2.2 and 3.1.1) can also be discussed in terms of plausibility. Features selected by this
method favor a generalizable model. In other words, discarded features may have some connection to ozone – but even if they





help to characterize the locations, their addition to the training data did not lead to a more generalizable model. This can have different reasons. As such, 'urban and built-up in 25 km area' was not selected presumably because urban areas are often very localized. Urban landcover in the area of 25 km around a location is therefore not as meaningful as the variables 'nightlight' and 'population density', which are like 'urban and built-up in 25 km area' proxies for human activity, but are available at higher resolution. Similarly, the feature 'cropland / natural vegetation mosaic in 25 km area' was discarded because ozone is affected differently by croplands and natural vegetation. Together with the large area considered, this feature becomes obsolete. We suspect the features 'snow and ice in 25 km area', 'barren or sparsely vegetated in 25 km area', and 'wheat production' did not contribute to the model generalizability because they are simply not represented well in the training data. A feature may be an important proxy for ozone, but if the relationship is not expressed in the training data, it cannot be learned by a machine learning model. This feature can become more important if other training locations are considered. This shows that the placing of measurement locations is crucial.

### 4.3   Mapping the global domain

For the global mapping, the model has to generalize to unseen locations. Two prerequisites are: 1) The model must have seen the feature combination during training. 2) The connection between features and the target, ozone, must be the same. The two conditions are only fulfilled in a very strictly constrained space, as can be seen in Fig. 7. Regarding the feature combinations, we combined cross validation with an inspection of the feature space as described in Sect. 2.3.1. Then, based on the cross validation on different world regions (Table 5), we decided that because the model uncertainty rises when training and testing in different world regions, we also combined cross validation in the spatial domain in Sect. 3.2.1. One interesting thing here to mention is that we conducted the same cross validation approach on other world regions with a shallow neural network (as in the baseline experiments of Betancourt et al. (2021b)). The neural network had similar evaluation scores on the test set, but it did not generalize as well to other world regions, showing even negative $R^2$ values when testing in other world regions (not shown). One reason for that could be that the random forest is an ensemble model and thus generalizes better under noisy data. We, therefore, decided to discard the neural network architecture, because our main goal is global generalizability.

Concerning our mapping approach, we can confidently map Europe, large parts of the US and East Asia, where the bulk of the measurement stations are located. Those are all industrialized countries in the northern hemisphere. Our cross validation results (Sect. 3.1.2), the area of applicability (Sect. 3.2.1), but also expert knowledge would agree that it is problematic to map to other world regions with the AQ-Bench training dataset only. However, the cross validation in connection with the area of applicability technique yielded also the knowledge that the models are not completely useless in other world regions. That is promising for future global mapping approaches. One idea to solve these problems of different connections between features and ozone in different world regions is to train localized models, and apply them wherever possible. Localized models could not only yield more accurate predictions but in connection with SHAP values (Sect. 2.2.4), they could also rule out the governing factors of ozone in the respective regions and be easier to interpret.

With regard to the spatial domain, we can also discuss the resolution. The model was trained on point data of the 'absolute latitude', 'altitude', and 'relative altitude', and technically one could produce more fine-grained maps if the input data is present





in higher resolution. The model is 'perfect' in this regard – because it was trained on infinite resolution point data as provided by TOAR. However, one may need to reconsider some assumptions made here in terms of regional representativity of the measurements and the relation between geographic features and ozone on a different scale.

### 4.4 Prospects for ozone mapping

In this study, we mapped average tropospheric ozone from the stations in the AQ-Bench dataset to a global domain. For this, we fused different auxiliary geospatial datasets and gridded data with machine learning. We chose to use features that are known to have a connection to ozone, and that were already proven to enable a prediction of ozone concentrations (Betancourt et al., 2021b). Our choice of data and algorithms is well justified and transparent. Errors did not exceed 5 ppb, which is also in the range of measurement error and therefore an acceptable uncertainty. The $R^2$ value of the final model is 0.55, which is a good value for properly validated mapping. The maps produced show known patterns of ozone such as lower levels in metropolitan areas and higher levels in mediterranean or mountainous regions. But there are situations – especially extremes (Fig. 10) – which are not predicted well. This can be considered as a general problem of machine learning (Guth and Sapsis, 2019) but was also noted in other ozone modeling studies (Young et al., 2018).

For this first approach, we limited ourselves to the static mapping of aggregated mean ozone. An advantage of this approach is that the model result is directly the ozone metric of interest (in this case average ozone). Since the AQ-Bench dataset contains other ozone metrics, they could be mapped as well. For example, vegetation- or health-related ozone metrics can be mapped with the same workflow and training dataset as described here. Another advantage is that we used a multitude of inputs that could not be used in a traditional model because their connection to ozone is unknown. This means we exploit two benefits of machine learning: first, obtaining a bias-free estimate of the target directly, and second, using a multitude of inputs with unknown direct impact on the target.

Our data product is a map that is aggregated in time. This could be a limitation as sometimes the data product of interest is a seasonal aggregate or even maps of daily or hourly air pollutant concentrations. In that regard, it is worth mentioning that the use of meteorological data in not-aggregated or aggregated form can be beneficial to further increase model performance. We applied a completely data-driven approach, relying heavily on geospatial data. The other side of the spectrum is DeLang et al. (2021), who fused chemical transport model output to observations without exploiting the connection to any auxiliary data. A possible direction to go from here is described by Irrgang et al. (2021), who propose the fusion of models and machine learning to benefit from both methods.

### 5 Conclusions

In this study, we developed a completely data-driven, machine learning-based, global mapping approach for tropospheric ozone. We mapped from the 5577 irregularly placed measurement stations of the AQ-Bench dataset (Betancourt et al., 2021b) to a regular $0.1° \times 0.1°$ grid. As environmental data, i.e. input features, we used a multitude of geospatial datasets. To our knowledge, this is the first completely data-driven approach to global ozone mapping. We combined this mapping with an



end-to-end approach for explainable machine learning and uncertainty estimation. This allowed us to assess the robustness,
525  scientific consistency, and global applicability of the model. We linked interpretation tools with domain knowledge to obtain
application-specific explanations, which is in line with Roscher et al. (2020). The methods are interconnected, e.g. forward
feature selection also made the model easier to interpret. Likewise, the area of applicability was shown to match the model's
robustness. We justified the choice of tools and detailed how the tools we have chosen provided us with the results we need to
make a comprehensive global analysis. The combination of explainable machine learning and uncertainty quantification makes
530  the model and outputs trustworthy. Therefore, the map we produced provides information on global ozone distribution and is a
transparent and reliable data product.

We explained the outcome and the model, which can lead to new scientific insights. Mapping studies like this one could
also contribute to studies like Sofen et al. (2016), that propose locations for new air quality measurement sites to extend the
observation network, to cover not only spatial world regions but also air quality regimes and areas with diverse geographic
535  characteristics. The approach of an area of applicability can also be used to decide where to build new measurement stations
to maximize the mapped area. The map as a data product can also be used to refine studies like TOAR (Fleming et al., 2018;
Mills et al., 2018) because it enables analyses at locations with no measurement stations. Closing the gaps in the maps, it would
be highly beneficial to also add station data from other countries, e.g. new data from East Asian countries, or from new data
sources such as OpenAQ[1].

540  *Code and data availability.* The mapping code which was used to generate the results published here is available under https://gitlab.jsc.
fz-juelich.de/esde/machine-learning/ozone-mapping (last access: 13 December 2021) under MIT License. The AQ-Bench dataset (Betan-
court et al., 2021b) is available under the DOI http://doi.org/10.23728/b2share.30d42b5a87344e82855a486bf2123e9f. The gridded data is
available under the DOI http://doi.org/10.23728/b2share.9e88bc269c4f4dbc95b3c3b7f3e8512c. The data products generated in this study,
namely the ozone map and the area of applicability are available under the DOI http://doi.org/10.23728/b2share.a05f33b5527f408a99faeaeea033fcdc.

---

[1]https://openaq.org/, last access 02 November 2021





545 **Appendix A: Technical details on the data**

Table A1: Technical details on the data used in this work. For more information on the station location data, refer to Betancourt et al. (2021b). Please note that 'land use in 25 km area' comprises all the different land cover features.

Table A1

| Variable | Data source and technical info | Reference |
|---|---|---|
| Ozone average values | Aggregated average ozone measurements of the stations in the AQ-Bench dataset from the years 2010-2014. The original data source is the database of the Tropospheric Ozone Assessment Report (TOAR). | Betancourt et al. (2021b), Schultz et al. (2017) |
| Climatic zone | Twelve classes of the IPCC 2006 classification scheme for default climate regions with a resolution of 5′. Stations were attributed to the climatic zone in the respective grid cell. To prepare the gridded field, downscaling to 0.1° resolution was done by nearest neighbor interpolation. | https://esdac.jrc.ec.europa.eu/ projects/RenewableEnergy/, accessed 23 Mar 2021 |
| Geographic location | The geographical location of the stations (longitude and latitude) was reported by the data providers and quality controlled by the TOAR database administrators. A gridded field of 0.1° resolution was generated within this study. | Schultz et al. (2017) |
| Altitude | The station altitude was reported by the data providers and quality controlled by the TOAR database administrators. The gridded field of 0.1° resolution was produced by linear 2D interpolation of the ETOPO 1 digital elevation model with an original resolution of 1′. | Schultz et al. (2017), Amante and Eakins (2009) |
| Relative altitude | Derived at stations from the ETOPO 1 digital elevation model and the station altitude. To generate a gridded field, the relative altitude was determined for every pixel from ETOPO 1 data. | Amante and Eakins (2009) |

*(continued on next page)*



*(Table A1 continued from previous page)*

| Variable | Data source | Reference |
|---|---|---|
| Land cover in 25 km area | Derived from yearly land cover type L3 from the MODIS MD12C1 collection with an original resolution of 0.05°. The year 2012 and the IGBP classification scheme with 17 classes were used. For the data at station locations, land cover data in the area of 25 km around each station was considered. Similarly, for the gridded fields, the 25 km area around each pixel was considered. | https://ladsweb.modaps.eosdis.nasa.gov/missions-and-measurements/products/MCD12C1/, accessed 23 Mar 2021 |
| Wheat / rice production | Annual wheat / rice production of the year 2000 according to the Global Agro-Ecological Zones data, version 3 with an original resolution of 5′. The stations were attributed with data of the respective pixel. The gridded field of 0.1° was produced by linear 2D interpolation. | www.fao.org/, accessed 23 Mar 2021 |
| $NO_x$ emissions | Annual $NO_x$ emissions of the year 2010 from EDGAR HTAP inventory V2 with an original resolution of 0.1°. The stations were attributed with data of the respective pixel. The gridded field of 0.1° was produced by linear 2D interpolation. | Janssens-Maenhout et al. (2015) |
| $NO_2$ full column | 5-year average (2011-2015) tropospheric $NO_2$ column value from the Ozone Monitoring Instrument (OMI) on NASA AURA with an original resolution of 0.1°. The stations were attributed with data of the respective pixel. | Krotkov et al. (2016) |
| Population density | GPWv3 population density of the year 2010 with an original resolution of 2.5′. For the data at station locations, data were aggregated in 1 km, 5 km, and 25 km around the station location. Similarly, for the gridded fields, data were aggregated in these radii around each pixel. | CIESIN (2005) |
| Nightlight | Stable nighttime lights of the year 2013 extracted from the NOAA DMSP product with an original resolution of 0.925 km. For the data at station locations, data were aggregated in 1 km, 5 km and 25 km around the station location. Similarly, for the gridded fields, data were aggregated in these radii around each pixel. | https://ngdc.noaa.gov/eog/dmsp/downloadV4composites.html, accessed 23 Mar 2021 |





## Appendix B: Plots of gridded fields used as inputs for mapping model

**Figure B1.** Gridded fields used for the final map production. Please note that the feature engineering was done as described in Sect. 2.2.2.





**Figure B2.** Gridded fields used for the final map production. Please note that the feature engineering was done as described in Sect. 2.2.2.





## Appendix C: Convergence of the error model

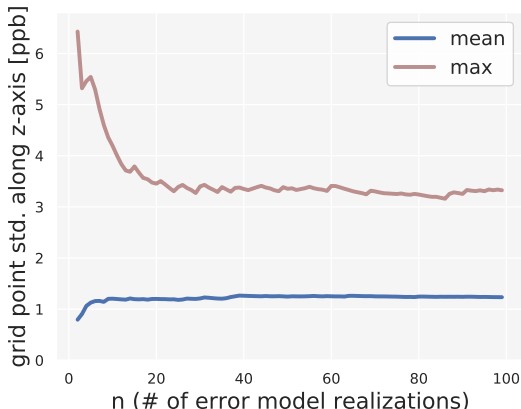

**Figure C1.** This plot justifies the use of 100 error model realizations in Sect. 3.2.2. We have stacked $n$ perturbed maps along the z axis. Then have monitored the grid point wise standard deviation along the z axis over these $n$ realizations of the error model. The mean standard deviation over the whole map stabilizes after ca. 40 realizations. The maximum standard deviation has some really high values for less than 20 realizations. This can be explained by the fact that for a low number of realiations, some grid points base their predictions on single, very differently perturbed stations. But this effect smoothes out after 20 realizations. Even though the maximum is not as stable as the mean (which is expected), convergence can be assumend after 100 realizations.



*Author contributions.* All authors jointly developed the concept of the project under the lead of CB and MGS. CB and SS coordinated the project. MGS, RR, and JK supervised the project. CB, TTS, AE, AP, and SS developed the code, conducted the experiments, and prepared the initial manuscript draft. MGS, RR, and JK reviewed and edited the manuscript. All authors read and approved the manuscript.

*Competing interests.* Martin G. Schultz is a topic editor of *Earth System Science Data* (ESSD) for the special issue "Benchmark datasets and machine learning algorithms for Earth system science data (ESSD/GMD inter-journal SI)" .

*Disclaimer.* Parts of this research were presented in oral and display format at the conference "EGU General Assembly 2021" (Betancourt et al., 2021a).

*Acknowledgements.* We are thankful to the TOAR community and several international agencies and institutions for making air quality and geospatial data available. We thank Hanna Meyer and Hu Zhao for helpful discussions. CB and SS acknowledge funding from the European Research Council, H2020 Research Infrastructures (IntelliAQ (grant no. ERC-2017-ADG#787576)). TTS, AE, AP, and SS acknowledge funding from the German Federal Ministry for the Environment, Nature Conservation and Nuclear Safety under grant no 67KI2043 (KISTE). RR acknowledges funding by the German Federal Ministry of Education and Research (BMBF) in the framework of the international future AI lab "AI4EO – Artificial Intelligence for Earth Observation: Reasoning, Uncertainties, Ethics and Beyond" (Grant number: 01DD20001). The authors gratefully acknowledge the Earth System Modelling Project (ESM) for funding this work by providing computing time on the ESM partition of the supercomputer JUWELS (Krause, 2019) at the Jülich Supercomputing Centre (JSC).



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
