# Peer review of "Global, high-resolution mapping of tropospheric ozone – explainable machine learning and impact of uncertainties"

_Geoscientific Model Development, 2022_

## Author Comment (AC2)

**Point-by-point response, Reviewer 1**

I appreciate the effort that the authors have put in assessing spatial uncertainties associated with their predictions. Unfortunately, this is often not a priority in global mapping papers, so I welcome this focus on uncertainty very much.

**Answer:** We thank reviewer 1 for the positive view of our work and appreciate the suggestions to improve the manuscript. The color 'light blue' denotes changes we made according to the suggestions of reviewer 1. Please note that all section and line numbers refer to the mark-up version of the manuscript. Figure numberings are shifted as we moved one figure from the end to the beginning of the paper as recommended by Reviewer 2.

The manuscript is structured well, but overall fairly lengthy and is written in a relaxed, almost conversational writing style - which I like, but from sometimes it's perhaps a bit too much. Another suggestion would be to move most of the tables and figures to the supplemental to improve readability. With 12 figures and 5 tables in the main text, I found it sometimes hard to navigate and find the most relevant results.

**Answer**: We agree that the original manuscript is quite long.

**Correction**: We improved the writing style and removed redundant information. This resulted in many small changes that are infeasible to list here. All changes are noted in the mark-up version of the manuscript. We moved Figs. 5 and 7 to annexes C and E, respectively. As a result, the manuscript was shortened by three pages.

I must admit that I know close to nothing about ozone and what variables structure its spatial patterns in the troposphere, but essentially the model is predicting O3 using latitude, altitude and human development (nearby nightlight); collectively explaining the majority of variation in the model. However, latitude and altitude are both merely a proxy for temperature and radiation; which are probably the actual main drivers of O3 levels (L153). Why not use them directly as predictors in the model? There are various high quality global layers available.

**Answer:** The purpose of this study is to map ozone using only static geospatial features and to evaluate the reliability of maps produced with these features. We recommend using meteorological data for future ozone mapping studies.

**Correction**: We now mention the use of static input features up-front in the introduction, line 72f: "This study builds upon Betancourt et al. (2021b) who proved that ozone metrics can be predicted using static geospatial data." We discuss the aspect of meteorological data use for future studies in Sect. 4.4, line 569f: "The use of meteorological data as static or non-static inputs can be beneficial to further increase model performance and allow time-resolved mapping."

With the very strong effect of absolute latitude in the model (~25% global importance), the predicted pattern will strongly reflect absolute latitude – which is clearly visible in the final map (fig 11).

**Answer**: We agree and found the latitude dependency consistent with our expectations. This property of the map was mentioned in the Visual analysis section (3.3.2).

**Correction**: We rephrased the sentence to be more explicit: "The global importance of 'absolute latitude' shows through a latitudinal stratification and a clear north-south gradient in Europe, the US and East Asia.", Sect. 3.3.2, line 431f. We also discussed this property of our map in more detail in Sect. 4.2, line 487ff: "Ozone is affected by meteorology (temperature, radiation) and precursor emissions (Sect. 1). The fact that there is no continuous increase of ozone towards tropical latitudes shows that the mapping model at least qualitatively captures the influence of low precursor emissions in the tropics. The importance of 'absolute latitude' also indicates that the model can be improved by including temperature and radiation features from meteorological data.".

Next, I have a couple of doubts with the current "area of applicability" (AOA) approach, where features are weighted by their respective importances as in the model. By weighting the features, you're essentially using absolute latitude and altitude (twice, including relative altitude) as to define feature space in which you apply the model. But as above; it's not latitude or altitude that define this space, but temperature and radiation (L153). A major drawback, I think, from scaling the features used in the AOA analysis using their respective importances, is that you're now basically assuming you can predict everywhere on the same latitude, save for some places (some of) the fewer important features fall outside the sampled space.

**Answer**: Scaling features according to their global importance is necessary as this global importance determines the features on which the model is basing its predictions. Basing the applicability of a random forest on features it seldomly uses to make a decision would be difficult to justify. Furthermore, not scaling the features at all would make correlated features multiple times important for applicability through the course of dimensionality. For example, the 'nightlight' in different radii around a station would have triple the importance of 'absolute latitude'. We agree with Meyer et Pebesma (2021) who state that the scaling 'lacks a formal statistical argument'. Nevertheless, our analysis shows that the unmodified method remains the best solution for this study.

Even though 'absolute latitude', 'altitude', 'relative altitude', and 'nightlight 5km' are most important, the other features are not ignored when defining the applicability. For example, large parts of the southern hemisphere are not predictable, even if they have exactly the same 'absolute latitude' as the northern hemisphere. Of course, the area of applicability might look different if other features would be used to train the model.

It would be useful if the authors would include a comparison with previously published tropospheric ozone predictions, based on mechanistic models. Of course, the authors did do a substantial test of their model's validity and stability – but these really only hold true for this particular model and dataset; and don't provide insight in how the results compare to other tropospheric ozone predictions.

**Answer**: The focus of this paper is the explainability and the exploration of mapping with static geospatial input features. We agree that a comparison with model data products is beneficial. We found the maps generated by DeLang et al. (2021) to be the most suitable, as they cover the same spatial domain and also aggregate in time. The Visual analysis Section (3.3.2) describes several spatial ozone patterns that can be found in both maps. We decided against a thorough quantitative analysis of the differences between the two maps, as this would go beyond the scope of this already rather long paper.

**Correction**: We now mention the similar patterns more explicitly in the scope of the visual analysis (Sect. 3.3.2, line 442f): "The spatial ozone patterns described here can also be found in ozone maps generated by traditional chemical models such as the fusion products by DeLang et al. (2021).".

Further points

1. The statement on L440 (*"The effect of 'absolute latitude' on predictions is consistent with what is known about ozone formation – ozone generally increases toward the equator"*) seems to be the opposite of the patterns that are predicted (Fig 11), with lower values near the equator, and higher values in temperate regions?

**Answer**: We agree this is misleading. The statement is only correct for the US, Europe, and Asia, not for the equator region itself. Ozone values are low in the equator regions due to a lack of precursor emissions.

**Correction**: We reformulated two sentences: Sect. 3.1.3, line 358f now reads "A lower 'absolute latitude' value leads to an increased ozone value prediction.". Sect. 4.2, line 483ff now reads "The effect of 'absolute latitude' on predictions is consistent with known ozone formation processes, i.e. ozone production generally increases where more sunlight is available. [...] Ozone is affected by meteorology (temperature, radiation) and precursor emissions (Sect. 1). The fact that there is no continuous increase of ozone towards tropical latitudes shows that the mapping model at least qualitatively captures the influence of low precursor emissions in the tropics.".

2. The selection of the threshold distance for 'non-predictable' data (ie., the upper whisker of all the cv distances), is seemingly arbitrary. It is in line with the AOA paper by Meyer and Pebesma, but neither that paper provides a statistical reasoning for picking this particular threshold.

**Answer**: The cross validation distances might contain outliers that would make the area of applicability unrealistically large. It is common to take the upper whisker as a robust threshold for outliers. We have tried the 95% percentile as an alternative to flag outliers with similar results.

3. The manuscript includes various subjective interpretations of the results. I believe the manuscript would benefit from a more objective wording. Some examples:

- L310: *"East Asia is a special case because the ozone value distribution is rather narrow there"*
- L325: *"Very high 'nightlight in 5km area' values"*

- L406: *"are considered significant"*
- Figure C1: *"really high values*

**Correction**: Thank you for pointing this out. We eliminated subjective language throughout the manuscript. Regarding the examples given by reviewer 1,

- We removed the statement in line 310.
- Line 359 now reads "High 'nightlight in 5km area' values lead to lower predicted ozone concentrations." Here we refer to 'high' as used in the color scale of Fig. 6: SHAP does not use absolute feature values but differentiates between high and low values.
- Line 450 now reads "We define a 5 ppb change in RMSE score or predicted ozone values as significant (Schultz et al. 2017)."
- The caption of Fig. D1 now reads "[…] The maximum standard deviation exceeds 3.5 ppb for less than 20 realizations. […]"

4. The authors do use "not shown" rather often, a total of 8 times in the entire manuscript. I guess this is ok, but if the authors feel that the data/results aren't necessary to show, arguable the entire section can be removed. If not, I would suggest placing the evidence for the statement in the supplemental materials.

**Correction**: We agree. We went through all eight occurrences of 'not shown'. We concluded that the information given was often irrelevant to the study. One instance of 'not shown' remains in line 348 because the similarity of the feature combinations of the cross validation sets is relevant for the study, but the exact values of the dissimilarity indices are not.

5. Figure 3: are these 'example data points' points in the AQ-bench dataset or in the raster data you're predicting?

**Answer**: These are data points we found in the gridded data.

**Correction**: We clarified this in the legend of Fig. 4.

---

## Author Comment (AC3)

**Point-by-point response, Reviewer 2**

The authors demonstrate a machine learning approach to generate high-resolution surface ozone concentration products, and evaluate the uncertainties from models and data sources. Many techniques are used in this study and they are generally explained well. The surface ozone products can be potentially used for other studies if the produced ozone mapping is robust. The manuscript is written well in a conversational way and I can feel that the authors try to add the novelty in explaining machine learning results, but overinterpreting should be avoided. There are a few major concerns that I think should be addressed carefully about the motivation of the study and the usage of final ozone products.

**Answer**: We thank reviewer 2 for the helpful suggestions to improve the manuscript. The color 'dark blue' denotes changes we made according to the suggestions of reviewer 2. Please note that all section and line numbers refer to the mark-up version of the manuscript. Figure numberings are shifted as we moved one figure from the end to the beginning of the paper as recommended by Reviewer 2.

Major comments:

1. High-resolution ozone mapping is a highlight of this study, but are there any differences between directly using interpolated original ozone products (TOAR) and the products generated here?

**Answer**: We decided against a direct interpolation for different reasons. Some studies compare direct interpolation from irregularly placed measurement sites to a regular grid, with mapping (Li et al. 2019, Ren et al., 2020 as cited in the manuscript). Mapping has better evaluation results in these studies. With a traditional interpolation method, the uncertainty of the map would be proportional to the distance of the measurement station. Also, the information inherent in the environmental features would not be used effectively. The factors on ozone formation change on very small regional scales. In contrast to geospatial interpolation techniques, the mapping approach can exploit similarities between distant sites such as precursor emission patterns. Furthermore, traditional interpolation techniques would not allow spatial extrapolation to regions without measurement stations. We could extend the map by a big margin compared to this approach (dark turquoise areas in Fig. 8).

**Correction**: We state these points more clearly in Sect. 1, line 49ff: "It was deployed for air pollution as an improvement over spatial interpolation and dispersion modeling which suffer from performance issues due to sparse measurements, and lack of detailed source description (Briggs et al., 1997).", line 55ff: "Several studies (e.g. Li et al., 2019; Ren et al., 2020), have shown that mapping using machine learning methods is superior to other geostatistical methods such as Kriging because it can capture nonlinear relationships and makes ideal use of environmental features by exploiting similarities between distant sites.", and line 58ff: "In contrast to traditional interpolation techniques, mapping allows to extend the domain to the global scale, because it can predict the variable of interest based on

environmental features, even in regions without measurements (Lary et al., 2014; Bastin et al., 2019; Hoogen et al., 2019)."

I think the authors try to extend the application to the regions where measurement sites are not available, but it is clearly that the trained results are limited to the number of measurement sites (mentioned in Sect. 3.2.1).

**Answer**: We showed that we could indeed cover large areas without measurements (dark turquoise areas in Fig. 8). Based on our cross validation (Sect. 3.1.2) and applicability analyses (Sect. 3.2.1), we pointed out that these areas suffer from higher RMSE than the locations with measurement coverage. The expected RMSE in the areas with measurements is still in the range of 5 ppb, so the produced map is also useful in these areas.

**Correction**: We state this more explicitly in the discussion, Sect. 4.3, line 522ff: "We combined cross validation with an inspection of the feature space to ensure matching feature combinations. Then, based on the cross validation on different world regions, we point out regions with sparse or no training data, where higher model errors are expected (Sect. 3.2.1)". Still in Sect. 4.3, we now state in line 536ff: "The cross validation results (Sect. 3.1.2), the area of applicability (Sect. 3.2.1), and expert knowledge confirm that uncertainties increase when a model trained on the AQ-Bench dataset is applied to other world regions. However, the cross validation in connection with the area of applicability technique shows that the model can be used in other world regions with acceptable uncertainties."

2. High-resolution ozone mapping may introduce extra uncertainties because input features or surface ozone concentrations may have large biases. Since surface ozone is regionally spread, slightly decreasing resolution may reduce uncertainties. This should be discussed to strengthen the motivation of the study.

**Answer**: The objective of this study is to obtain high resolution maps that cover small scale features. We chose the resolution of 0.1° because it can capture typical long-term ozone patterns on small scales. Evidence that this has worked well is provided by the significantly lower ozone levels in European metropolitan areas in Figs. 11 and 12. We do not aim for an improved signal to noise ratio through more aggregations.

3. The averaged surface ozone concentrations over 2004-2014 are reproduced and the authors also mentioned that the products are static in Sect. 4.4. The geographic variables are used to drive the machine learning model and many of them (e.g. latitude, altitude) instead of physical or chemical variables show high importance to simulated ozone. The relationships between some variables are intuitive, but the issue is that simulated future ozone concentrations may be quite similar to those simulated over 2004-2014 because the geographic variables with high importance are static – they will not change in the future. The temporal relationships may not be captured by the machine learning model.

**Answer**: It is not the intention of this study to forecast ozone, nor to project to years the model was not trained on. We agree that most of the most important variables ('absolute latitude', 'altitude', 'relative

altitude') do not change in the course of years, while 'population density' or 'nightlight' can indeed change. To project to different years, one would have to train on both ozone data and gridded data from these years, and a model would be only valid for the year it was trained on, as far as we know. Cross-validation for different years (as we have done for the different world regions) would be necessary to prove otherwise, but this is beyond the scope of this study. One would need to include meteorological data to improve predictions and to make them time resolved. We hope that changes in emission sources in time would be seen in proxies such as nightlights. Further testing would be necessary to show these factors in a machine learning model. This is an interesting research direction.

**Correction**: We clarified these limitations. Section 4.4., line 566f now reads: "Our model is only valid for the training data period (2010-2014), and it is not suitable to predict ozone values in other years."

It may be useful to justify the usage of average ozone earlier in the data description section, and to state the benefits of using final ozone products.

**Correction**: We agree. Section 2.1.1, line 89ff now reads: "The AQ-Bench dataset considers ozone concentrations on a climatological time scale instead of day-to-day air quality data. The scope of this dataset is to discover purely spatial relations. Machine learning models trained on this dataset will output aggregated statistics over the years 2010 - 2014, and will not be able to capture temporal variances. This is beneficial if the required final data products are also aggregated statistics.".

Other comments:

1. Line 11: "By inspecting the feature space, …". Not clear in the abstract.

**Correction**: We agree this might be unclear. The Abstract, line 11f now reads: "By inspecting the input features, we ensure that the model is only applied in regions where it is reliable.".

2. Line 59: Need to clearly point out the key issues in the current mapping field and what the benefits are by using machine learning approaches.

**Correction**: We now explain the key issues in more detail. Sect. 1, line 62ff reads: "Meyer et al. (2018) and Ploton et al. (2020) point out that some studies may be overconfident because they validate their maps on data that is not statistically independent from the training data. This occurs when a random data split is used on data with spatio-temporal (auto)correlations. There are also concerns when the mapping models are applied to areas that have completely different properties from the measurement locations (Meyer and Pebesma, 2021). A model trained on certain input feature combinations can only be applied to similar feature combinations."

3. Line 79: Please justify the usage of annual mean surface ozone concentrations. I suppose that using monthly data would make the model more robust as more data are involved in the training?

**Answer**: We do not use annual means, but aggregates over the years 2010-2014 because we aim for a prediction on the climatological time scale. More training data would make the model more robust, but we cannot combine time resolved ozone concentrations with static input features. To explain monthly variations, we would need monthly resolved input data, such as meteorological data. This is the next step we would like to take in the future.

**Correction**: We refer again to Section 2.1.1, line 89ff, where we now explain in more detail why we use temporal aggregates. The prospects of time resolved mapping are mentioned in the discussion (Sect. 4.4, line 569f), and we now added it to the conclusion (Sect. 5, line 595ff): "It would be beneficial to add time resolved input features to the training data to improve evaluation scores and increase the temporal resolution of the map. Adding training data from regions like East Asia, or new data sources such as OpenAQ would close the gaps in the global ozone map."

4. Line 76: Are there only 5577 data used for machine learning?

**Answer**: Yes, this is correct. These are the stations in the TOAR database that have sufficient data capture in the years 2010-2014. We state N=5577 in Fig.1 as part of the basic statistics.

5. A more specific title is needed for Figure 1 instead of saying 'average ozone values'.

**Correction**: The caption of Fig. 1 now reads: "Average ozone statistic of the AQ-Bench dataset. The values at 5577 measurement stations are aggregated over the years 2010-2014. (a) Values on a map projection. (b) Histogram and summary statistics."

6. Line 86: I am not convinced by the association between 'latitude' and ozone photochemistry.

**Correction**: We explain the association in more detail. Sect. 2.1.1, line 98f now read "'Latitude' is a proxy for ozone formation through photochemistry, as radiation and heat generally increase towards the equator.". We discuss the scientific consistency of this relationship in more detail in Sect. 4.2, line 487ff: "Ozone is affected by meteorology (temperature, radiation) and precursor emissions (Sect. 1). The fact that there is no continuous increase of ozone towards tropical latitudes shows that the mapping model at least qualitatively captures the influence of low precursor emissions in the tropics. The importance of 'absolute latitude' also indicates that the model can be improved by including temperature and radiation features from meteorological data."

7. In Table. 1, many land cover variables are used so they may principally reflect ozone dry deposition? Some discussions are needed here.

**Correction**: Yes. We clarified this in line 99f: "The landcover variables are proxies for precursor emissions and deposition.". More details are available in Betancourt et al. (2021b) as cited in the manuscript.

8. NOx emissions and columns are used. What about other ozone precursor emissions?

**Answer**: The other ozone precursors are reflected in land cover (VOCs), population density (engine exhaust, CO), and nightlights (industrial or human activities, CO). These proxies cover all known ozone precursors. Section 2.1.1 briefly mentions these precursor sources without going into too much detail. A full description of the proxies and pathways to ozone formation is beyond the scope of this study. The interested reader should refer to Betancourt et al. (2021b), cited in this manuscript.

9. Line 106: It is too confident to state that the random forest is the most suitable; apparently, it is not.

**Correction**: The Sect. 1, line 125f now reads: "In addition, this algorithm has been proven to be suitable for mapping in several studies (Petermann et al., 2021; Nussbaum et al., 2018; Ren et al., 2020).".

10. Figure 3: Does the data points outside the area of applicability simply mean they have extreme high or low values that are not easily to predict?

**Answer**: The red points are example data points found in the gridded data. They have a large distance in the feature space to the AQ-Bench cluster. It is irrelevant if the feature values are very high or very low, they are simply in a location of the feature space that is not covered with training data. Therefore, a model trained on the AQ-Bench dataset cannot predict the red points.

**Correction**: We clarified the legend of Figure 4, the description of the red points now reads: "Example gridded data points outside area of applicability"

As you scale feature values with SHAP values, it is likely that the threshold used to filter large values is largely dependent on altitude.

**Answer**: This is to some extent correct and intended. Not scaling the features at all would make correlated features multiple times important for applicability through the course of dimensionality. For example, the 'nightlight' in different radii around a station would have triple the importance of 'absolute latitude'.

The threshold is not determined by a single feature, but by the combination of features. We refer to Fig. 4, where it is clearly visible that depending on the 'absolute latitude', a smaller or larger variety of '(relative) altitude' is covered by the training data. It is correct that altitudes play an important role here, and this makes sense from an ozone researcher's perspective as well: There are few measurement stations in high altitudes, most of which are situated in densely populated valleys. The model is therefore unsuitable to make predictions on high mountains.

11. Line 303: I cannot judge if RMSEs in the range of 3.84 to 4.04 ppb are large or small, even though the authors said this is acceptable. I think it will be better to show temporal one standard deviation of surface ozone concentrations along with surface ozone mixing ratios (annual mean) in Fig. 1 for readers.

**Answer**: Thank you for pointing this out. Fig. 1 shows the standard deviation of 6.40 ppb.

**Correction**: Sect. 3.1.2, line 336ff now reads: "Putting this RMSE value into perspective, 5 ppb is a conservative estimate for the ozone measurement error (Schultz et al., 2017). It is also lower than the 6.40 ppb standard deviation of the true ozone values of the training dataset (Fig. 1)."

12. Line 327: SHAP value discussions are in Sect. 4.2. I suggest that the authors avoid using many forward references, and merge some discussions in the corresponding sections.

**Answer**: We chose to separate the results (Sect. 3) from the discussions (Sect. 4). We believe that the forward references are necessary because the results from Sect. 3 are grouped by discussion topic in Sect. 4.

13. The evaluation picture (Fig. 10) is important, and I suggest to move it forward.

**Correction**: We agree. The figure is now appearing in the data description section (Sect. 2.1.1, reference in line 101f).

14. Two panels should be indicated in Fig. 11. It would be interesting to show the readers the predicted surface ozone mixing ratios across the globe, even if the authors identify some areas as inapplicable.

**Answer**: We decided against presenting these values because they have a bias that we cannot quantify. We have also removed these values from the map as a data product because we are concerned that others may reuse these data, although we do not recommend this.

15. Line 445: I think this is overinterpreted as you are using nightlight conditions to explain monthly or annual mean ozone variation. Ozone chemical production or destruction depends on NOx concentrations and NO titration is one aspect. It is fine if some relationships cannot be explained and I don't expect the relationships derived from SHAP values can explain every feature because machine learning model is not process-based.

**Answer**: Correct, the model does not learn the process of NOx titration. Nevertheless, it can learn the effects of processes without being process based, as they are visible in the data.

**Correction**: We clarify this in Sect. 4.2, line 501f: "It is also not expected that the machine learning model learns the ozone related processes described above because it is not process based. Instead, it learns the effects of processes if they are reflected in the training data."

16. How do authors think of the relative importance of training data number and training strategies (e.g. model types, feature selections) in ozone mapping? The number of training data may be more important shown in the study, and there is a need to discuss this aspect.

**Answer**: Based on our analyses we found model types and feature selection methods not to influence the model performance. Concerning adding data, there are different options. a) Adding more data from the training regions proved not to be effective: Training on 60% versus 80% of the AQ-Bench dataset did not enhance the model performance. This is presumably because the task posed (predicting average ozone based on static geospatial features) does not allow for better performance metrics. b) Adding data from regions with different characteristics would allow the map to cover more regions, but would not enhance the performance metrics. c) We expect that adding meteorological data or time resolved data would be beneficial for the model performance, and we now state that in the conclusions.

**Correction**: We added this aspect to the conclusions. Sect. 5, line 587ff now reads: "Mapping studies like this one could also contribute to studies like Sofen et al. (2016), that propose locations for new air quality measurement sites to extend the observation network. Here the inspection of the feature space helps to cover not only spatial world regions but also air quality regimes and areas with diverse geographic characteristics. Building locations can also be proposed based on their contribution to maximizing the area of applicability (Stadtler et al., 2022). [...] It would be beneficial to add time resolved input features to the training data to improve evaluation scores and increase the temporal resolution of the map. Adding training data from regions like East Asia, or new data sources such as OpenAQ would close the gaps in the global ozone map."

---

## Author Response (AR2)

Dear Peter Düben and topical editors of the Geoscientific Model Development Journal,

Thank you for accepting our manuscript "Global, high-resolution mapping of tropospheric ozone – explainable machine learning and impact of uncertainties" for publication.

Please note that I have added two statements to the acknowledgments in this version. The requirements of acknowledging DFG have changed, and we would like to thank the anonymous reviewers:

"[…] Open Access publication funded by the Deutsche Forschungsgemeinschaft (DFG, German Research Foundation) – 491111487. We thank two anonymous reviewers for their suggestions to improve this work."

Kind regards,

Clara Betancourt